# RIPK1 Regulates Microglial Activation in Lipopolysaccharide-Induced Neuroinflammation and MPTP-Induced Parkinson’s Disease Mouse Models

**DOI:** 10.3390/cells12030417

**Published:** 2023-01-26

**Authors:** Do-Yeon Kim, Yea-Hyun Leem, Jin-Sun Park, Jung-Eun Park, Jae-Min Park, Jihee Lee Kang, Hee-Sun Kim

**Affiliations:** 1Department of Molecular Medicine and Inflammation-Cancer Microenvironment Research Center, School of Medicine, Ewha Womans University, Seoul 07804, Republic of Korea; 2Department of Physiology and Inflammation-Cancer Microenvironment Research Center, School of Medicine, Ewha Womans University, Seoul 07804, Republic of Korea

**Keywords:** RIPK1, necrostatin-1, microglial activation, neuroinflammation, necroptosis, Parkinson’s disease, neuroprotection

## Abstract

Increasing evidence suggests a pivotal role of receptor-interacting protein kinase 1 (RIPK1), an initiator of necroptosis, in neuroinflammation. However, the precise role of RIPK1 in microglial activation remains unclear. In the present study, we explored the role of RIPK1 in lipopolysaccharide (LPS)-induced neuroinflammation and 1-methyl-4-phenyl-1,2,3,6-tetrahydropyridine (MPTP)-induced PD model mice by using RIPK1-specific inhibitors necrostatin-1 (Nec-1) and necrostatin-1 stable (Nec-1s). Nec-1/Nec-1s or RIPK1 siRNA inhibited the production of proinflammatory molecules and the phosphorylation of RIPK1-RIPK3-MLKL and cell death in LPS-induced inflammatory or LPS/QVD/BV6-induced necroptotic conditions of BV2 microglial cells. Detailed mechanistic studies showed that Nec-1/Nec-1s exerted anti-inflammatory effects by modulating AMPK, PI3K/Akt, MAPKs, and NF-κB signaling pathways in LPS-stimulated BV2 cells. Subsequent in vivo studies showed that Nec-1/Nec-1s inhibited microglial activation and proinflammatory gene expression by inhibiting the RIPK1 phosphorylation in the brains of LPS-injected mice. Furthermore, Nec-1/Nec-1s exert neuroprotective and anti-inflammatory effects in MPTP-induced PD mice. We found that p-RIPK1 is mainly expressed in microglia, and thus RIPK1 may contribute to neuroinflammation and subsequent cell death of dopaminergic neurons in MPTP-induced PD model mice. These data suggest that RIPK1 is a key regulator of microglial activation in LPS-induced neuroinflammation and MPTP-induced PD mice.

## 1. Introduction

Neuroinflammation plays a key role in the pathogenesis of neurodegenerative diseases [1,2]. Microglia are major cells involved in neuroinflammation and can be beneficial or detrimental depending on their phenotypes. In their resting state, microglia produce neurotrophic factors, support neuronal survival, and remove cell debris [3]. When microglia are activated by brain injury, microbial infections, or aggregated proteins, they release proinflammatory and neurotoxic factors and aggravate neurodegeneration and neuronal cell death [4,5]. Parkinson’s disease (PD) is the second most common neurodegenerative disease after Alzheimer’s disease (AD). The major pathogenesis factors of PD are selective loss of dopaminergic neurons in substantia nigra (SN), α-synuclein aggregation termed Lewy bodies, and neuroinflammation [5,6,7]. Recently, necroptosis was shown to be involved in cell death, inflammation, and protein aggregation, implying that necroptosis may play a pivotal role in the pathogenesis of PD [8,9].

Necroptosis is programmed necrosis, which is morphologically similar to necrosis but genetically programmed/regulated, similar to apoptosis, and causes inflammation [8,10]. Necroptosis is induced by toll-like receptor (TLR) agonists, ligands of death receptors, microbial infections, and receptor-interacting protein kinase 1 (RIPK1); RIPK3 and its substrate mixed lineage kinase domain-like protein (MLKL) are known to play an important role in the process [11]. The role of necroptotic factors has been best studied by focusing on TNF receptor 1 (TNFR1) [12]. When TNF-α binds to TNFR1, complex I is formed in the cell, and RIPK1 is ubiquitinated by an E3 ligase, such as, for example, cellular inhibitor of apoptosis protein 1 (cIAP1). The ubiquitinated RIPK1 acts as a docking site for NF-κB essential modulator (NEMO; IKK-γ) and subsequently activates NF-κB, thereby inducing inflammation. Cylindromatosis (CYLD) deubiquitinates RIPK1, dissociates complex I, and induces the formation of complex IIa or complex IIb. Complex IIa comprises FAS-associated death domain protein (FADD), caspase 8, and RIPK1; it induces apoptosis by activating caspase 8 and its downstream effector, caspase 3. In this situation, RIPK1 is degraded by caspase 8. When caspase 8 activity is inhibited, RIPK1 is activated and binds to RIPK3, forming complex IIb. Thereafter, MLKL, which was phosphorylated by RIPK3, oligomerizes and translocates to the plasma membrane and induces membrane rupture and necroptosis [13]. 

RIPK1, an initiator of necroptosis, has emerged as a key upstream regulator of cell death and inflammation [14,15]. The RIPK1 increase has been observed in the microglia of AD, amyotrophic lateral sclerosis (ALS), and multiple sclerosis (MS) mouse models, and the RIPK1 inhibitor necrostatin-1 (Nec-1) treatment reduced proinflammatory cytokine expression [16]. Nec-1 is the first small-molecule inhibitor of RIPK1, and Nec-1 stable (Nec-1s), also called 7-Cl-O-Nec-1, is a more stable optimized analog of Nec-1 with reduced off-target effects [11,17]. The neuroprotective effect of Nec1/Nec-1s has been reported in several neurodegenerative diseases [9]. The Nec-1/Nec-1s reduced infarct volume in cerebral ischemia [11]. Nec-1s reduced amyloidosis, Aβ levels, and behavioral deficits in AD model mice [18]. Nec-1s are also protected from demyelination by preventing the TNF-α-mediated necroptosis of oligodendrocytes in MS models [19]. Recently, the neuroprotective effects of Nec-1/Nec-1s have been reported in both in vitro and in vivo PD models [20,21,22,23]. However, the detailed mechanisms underlying the effect of Nec-1/Nec-1s on neuroinflammation and neuroprotection in PD models have not been clearly demonstrated. 

In this study, we investigated whether Nec-1 and Nec-1s have therapeutic effects in lipopolysaccharide (LPS)-induced neuroinflammation and MPTP-induced PD mouse models. We showed that Nec-1 and Nec-1s exerted anti-inflammatory effects in LPS-induced neuroinflammatory/necroptotic conditions both in vitro and in vivo and analyzed the detailed molecular mechanisms. Furthermore, the neuroprotective and anti-inflammatory effects of Nec-1/Nec-1s were shown in MPTP-induced PD mice. Through these studies, we demonstrated that RIPK1 plays a role as a key regulator of microglial activation in neuroinflammation and PD mouse models.

## 2. Materials and Methods

### 2.1. Animals 

Male C57BL/6 mice (22–25 g, 7 weeks old) were purchased from Orient Bio Inc. (Seongnam, Korea). The mice were housed at 21 °C under a 12 h light:12 h dark cycle and had ad libitum access to food and water. Every effort was made to minimize the suffering of the animals. All experiments were performed in accordance with the National Institutes of Health and Ewha Womans University guidelines for the Care and Use of Laboratory Animals, and the study was approved by the Institutional Animal Care and Use Committee of the Medical School of Ewha Womans University (#EUM 21-012).

### 2.2. BV2 Microglial Cell Culture

The immortalized mouse BV2 microglial cell line [24] was grown and maintained in Dulbecco’s modified Eagle’s medium (DMEM) supplemented with 10% heat-inactivated fetal bovine serum (FBS), streptomycin (10 μg/mL), and penicillin (10 U/mL) at 37 °C in an incubator with 5% CO_2_.

### 2.3. Reagents and Antibodies

Nec-1, 7-Cl-O-Nec-1 (Nec-1s), and BV-6 (BV6) [Smac mimetic compound; antagonize IAP] were purchased from Selleck Chemicals (Houston, TX, USA). Q-VD-OPh (QVD; pan- caspase inhibitor; does not inhibit caspase-1 activity) [25] was purchased from R&D Systems (Minneapolis, MN, USA). LPS (strong immunostimulant from Escherichia coli serotype 055: B5) was obtained from Sigma-Aldrich (St. Louis, MO, USA). MPTP (dopaminergic neuron-specific toxin) was purchased from Tokyo Chemical Industry Co. (Tokyo, Japan). The following antibodies were purchased from Cell Signaling Technology (Danvers, MA, USA): Phospho- and total forms of mitogen-activated protein kinases (MAPKs), AMP-activated protein kinase (AMPK), Akt, cAMP response element-binding protein (CREB), phospho- and total forms of RIPK1, MLKL, total form of RIPK3, interleukin (IL)-6, TH. Antibodies against IL-10 and TNF-α were purchased from Santa Cruz Biotechnology (Dallas, TX, USA). Antibodies against heme oxygenase-1 (HO-1), Iba-1, and phospho-form of RIPK3 were purchased from Enzo Life Sciences (Farmingdale, NY, USA), Wako (Osaka, Japan), and Abcam (Waltham, MA, USA), respectively. All other chemicals were obtained from Sigma-Aldrich unless otherwise stated.

### 2.4. Measurement of Nitrite, Cytokine, and Intracellular Reactive Oxygen Species (ROS) Levels

For inflammatory conditions, BV2 cells (1 × 10^5^ cells per well in a 24-well plate) were pretreated with Nec-1, Nec-1s 1 h prior to LPS stimulation (100 ng/mL) for 8.5 h. For necroptosis conditions, BV2 cells were pretreated with Nec-1, Nec-1s (200 μM) 1 h prior to LPS stimulation (100 ng/mL) for 3 h and were then pretreated with QVD (10 μM) for 30 min before stimulation with BV6 (5 μM) for 5 h [25,26]. Supernatants of the cultured cells were collected, and nitrite accumulation was measured using Griess reagent (Promega, Madison, WI, USA). The concentrations of TNF-α, IL-1β, IL-6, and IL-10 were measured using an enzyme-linked immunosorbent assay (ELISA) with a kit supplied by BD Biosciences (San Jose, CA, USA). The intracellular accumulation of ROS was measured with H_2_DCF-DA (Sigma-Aldrich, St. Louis, MO, USA), as previously described [26].

### 2.5. Cytotoxicity Assay (LDH Assay)

Lactate dehydrogenase (LDH) release into media was measured using the CyQUANT LDH Cytotoxicity Assay kit according to the manufacturer’s instructions (Invitrogen, Waltham, MA, USA). The maximal LDH release levels were measured by treating the cells with 10× lysis solution to completely lyse the cell. Absorbance data were obtained using a 96-well plate reader (Molecular Devices, San Jose, CA, USA) at 490 nm and 680 nm. To determine LDH release (%), the absorbance at 680 nm was subtracted from that at 490 nm according to the manufacturer’s instructions.

### 2.6. Western Blot Analysis

Whole cell protein lysates and brain tissue homogenates were prepared in RIPA buffer consisting of 10 mM Tris (pH 7.4), 300 mM NaCl, 1% Triton X-100, 0.1% SDS, 0.1% sodium deoxycholate, 1 mM EDTA, and protease inhibitor cocktail. Protein samples were separated using SDS-PAGE, transferred to a nitrocellulose membrane, and incubated with primary antibodies diluted according to the manufacturer’s instructions. Specific details of antibodies used in Western blot analysis are summarized in Appendix A. After the membranes were thoroughly washed with TBST, HRP-conjugated secondary antibodies (BioRad, Hercules, CA, USA, 1:2000 dilution in 5% skim milk) were applied, and the blots were developed using an enhanced chemiluminescence detection kit (Thermo Fisher Scientific, Waltham, MA, USA). For quantification, the density of specific target bands was normalized against β-actin using ImageJ software, version 1.37 (National Institutes of Health, Bethesda, MD, USA).

### 2.7. Reverse-Transcription Polymerase Chain Reaction (RT-PCR)

Total RNA from BV2 cells and mouse brain tissue was extracted using TRIzol reagent (Invitrogen, Waltham, MA, USA). For RT-PCR, total RNA (1 μg) was reverse transcribed in a reaction mixture containing 500 ng random primers, 3 mM MgCl_2_, 0.5 mM dNTP, 1× RT buffer, and 10 U reverse transcriptase (Promega, Madison, WI, USA). The synthesized cDNA was used as a template for the PCR reaction using GoTaq polymerase (Promega, Madison, WI, USA) and primers. RT-PCR was carried out in a Bio-Rad T100 thermal cycler (Bio-Rad, Hercules, CA, USA). Quantitative RT-PCR was performed using a QuantStudio™ 3 Real-Time PCR System (Applied Biosystems, Waltham, MA, USA) with SYBR Green PCR Master Mix (Bioline, Memphis, TN, USA). The expression levels of target genes were normalized against that of GAPDH using the following formula: 2^(Ct, test gene − Ct, GAPDH)^. The primer sequences used in the PCR reactions are shown in Table 1.

### 2.8. Electrophoretic Mobility Shift Assay (EMSA)

BV2 cells were pretreated with Nec-1 or Nec-1s for 1 h and stimulated with LPS (100 ng/mL) for 1 h. The nuclear extracts from the cells were prepared as previously described [27]. Double-stranded DNA oligonucleotides containing the NF-κB and antioxidant response element (ARE) consensus sequence (Promega, Madison, WI, USA) were end-labeled using T4 polynucleotide kinase (New England Biolabs, Ipswich, MA, USA) in the presence of [γ-^32^P] ATP. Nuclear proteins (5 μg) were incubated with a ^32^P-labeled probe on ice for 30 min, resolved on a 5% acrylamide gel, and visualized by autoradiography.

### 2.9. Transient Transfection and Luciferase Assay

BV2 cells (2 × 10^5^ cells/well on a 12-well plate) were transfected with 1 μg of reporter plasmid DNA using Metafectene transfection reagent (Biontex, Martinsried/ Planegg, Germany). After 36 h of transfection, the cells were treated with LPS (100 ng/mL) for 7 h in the presence or absence of Nec-1. Then, the luciferase assay was performed to determine the effect of Nec-1 on reporter gene activity. siRNA-targeting mouse RIPK1 mRNA and scrambled control siRNA were obtained from Sigma-Aldrich (MISSION siRNA and MISSION siRNA Universal Negative Control, respectively). RIPK1 siRNA was transfected into BV2 cells using Metafectene transfection reagent according to the manufacturer’s protocols. The RIPK1 siRNA sequences were sense, 5′-GCCAAAUGUAUAGUACUUA-3′; antisense, 5′-UAAGUACUAUACAUUUGGC-3′.

### 2.10. Drug Administration

To study the LPS-induced systemic inflammation mouse model, C57BL/6 mice were randomly divided into five groups (Control, LPS, LPS+Nec-1, LPS+Nec-1s, Nec-1; each group, *n* = 8–10). Nec-1 or Nec-1s were dissolved in vehicle (1% DMSO in normal saline) and administered daily (100 mg/kg, intraperitoneal; i.p.) for four days. LPS (5 mg/kg, i.p.) was injected 1 h after the final Nec-1 administration as previously described [27]. For studying the MPTP-induced PD mouse model, C57BL/6 mice were divided into six groups (Control, MPTP, MPTP+Nec-1, MPTP+Nec-1s, Nec-1, and Nec-1s; each group, *n* = 12–14). Nec-1, Nec-1s (100 mg/kg, i.p.) was administered daily for three consecutive days. One day after the final Nec-1/Nec-1s treatment, mice were injected with MPTP (20 mg/kg, i.p.) four times at 2 h intervals, and mice were sacrificed after 7 days of MPTP injection [28].

### 2.11. Brain Tissue Preparation

For histological analysis, the mice were anesthetized with sodium pentobarbital (80 mg/kg body weight, i.p.) and then perfused transcardially with 0.9% saline followed by 4% paraformaldehyde for tissue fixation. The brains were then isolated and stored in 30% sucrose solution at 4 °C for cryoprotection. For biochemical analysis, the mice were transcardially perfused with saline. For histological analysis, serial coronal brain sections (40 μm thick) were cut using a cryostat (CM 1860; Leica, Wetzlar, Germany).

### 2.12. Immunohistochemistry and Immunofluorescence Analysis

For immunohistochemical (IHC) staining, sections were treated with 3% H_2_O_2_ and 4% BSA to inactivate endogenous peroxidation and block non-specific binding, respectively. The sections were incubated with a primary antibody against tyrosine hydroxylase (TH) or Iba-1 (1:1000) overnight, then with biotinylated secondary antibodies for 1 h at 25 °C room temperature, followed by an avidin-biotin-HRP complex reagent solution (Vector Laboratories, Burlingame, CA, USA). Subsequently, the peroxidase reaction was performed using diaminobenzidine tetrahydrochloride (Vector Laboratories, Burlingame, CA, USA). For double immunofluorescence (IF) staining, sections were treated to block non-specific binding and were incubated with primary antibodies, followed by secondary antibodies conjugated to a fluorophore. Specific details of antibodies used in IHC and IF staining are summarized in Appendix A. Digital images of the IF staining were captured using a Leica DM750 microscopy and quantified using Image J. Four to six sections per brain were stained and quantified. The number of target protein-positive cells per area (mm^2^) was counted, and the co-localization rate (%) was calculated using the following formula: {(the number of target protein-positive cells co-stained with cell type marker)/the number of cell type marker-positive cells} × 100.

### 2.13. Statistical Analysis

Differences between experimental groups were analyzed by one-way analysis of variance, and post-hoc comparisons were made using Tukey’s test. All statistical analyses were conducted using SPSS for Windows, version 18.0 (SPSS Inc., Chicago, IL, USA). In in vivo experiments, the sample size was not predetermined. All values are reported as mean ± standard error of mean (SEM). A *p*-value < 0.05 was considered statistically significant.

## 3. Results

### 3.1. Nec-1 and Nec-1s Showed Anti-Inflammatory Effects in LPS or LPS/QVD/BV6-Stimulated BV2 Microglial Cells

In this study, we examined the effect of RIPK1 inhibitor Nec-1 and Nec-1s in LPS-induced inflammatory conditions and LPS/QVD/BV6-induced necroptotic conditions. Necroptotic cell death was considered inflammatory in part due to cell lysis and the subsequent release of damage-associated molecular patterns (DAMPs), which can induce or amplify cytokine levels [29]. To induce necroptosis, BV2 cells were preincubated with LPS (100 ng/mL) for 3 h and were then pretreated with QVD (10 μM, pan-caspase inhibitor) for 30 min before stimulation with BV6 (5 μM, Smac mimetic; IAP inhibitor) for 5 h [25] (Figure 1A). On the other hand, BV2 cells were treated with LPS (100 ng/mL) only for 8.5 h to induce inflammation. We observed that almost the equivalent amount of NO, TNF-α, IL-6, and IL-10 was released into media by LPS or LPS/QVD/BV6 treatment (Figure 1B,C). However, IL-1β production was remarkably increased by LPS/QVD/BV6 treatment compared with LPS alone. We found that Nec-1 and Nec-1s suppressed the production of NO and pro-inflammatory cytokines TNF-α, IL-1β, and IL-6, while they increased the anti-inflammatory cytokine IL-10 in LPS-stimulated BV2 cells (Figure 1B). Similar results were observed in necroptotic microglia (Figure 1C).

### 3.2. Knockdown of RIPK1 Recapitulated the Anti-Inflammatory Effect of Nec-1/Nec-1s in LPS- or LPS/QVD/BV6-Stimulated BV2 Cells

To confirm whether the effects of Nec-1/Nec-1s are mediated through RIPK1 inhibition, we examined the effect of RIPK1 siRNA on pro-/anti-inflammatory molecules induced by LPS or LPS/QVD/BV6 (Figure 1D,E). As shown in Figure 1E, RIPK1 siRNA suppressed the production of NO, TNF-α, IL-1β, IL-6, and ROS, while IL-10 levels increased in LPS or LPS/QVD/BV6-stimulated BV2 cells, suggesting that the anti-inflammatory effects of Nec-1/Nec-1s are at least partly mediated through RIPK1 inhibition.

### 3.3. Nec-1 and Nec-1s Inhibited RIPK1-RIPK3-MLKL Phosphorylation and Cell Death in LPS or LPS/QVD/BV6-Stimulated BV2 Cells

Since Nec-1 and Nec-1s are known to block RIPK1-RIPK3-MLKL signal transduction by inhibiting RIPK1 phosphorylation [30], we examined the effect of Nec-1/Nec-1s on these three necroptotic mediators. We found that LPS induced the phosphorylation of RIPK1, RIPK3, MLKL, and the fold induction was remarkably increased upon LPS/QVD/BV6 treatment (Figure 2A,B). The total protein expression of MLKL was also increased by LPS and further by LPS/QVD/BV6, whereas total RIPK1 and RIPK3 levels were unchanged. Nec-1 and Nec-1s significantly inhibited the phosphorylation of RIPK1, RIPK3, and MLKL in LPS- or LPS/QVD/BV6-stimulated BV2 cells (Figure 2A,B). However, their total protein levels were unaltered by Nec-1/Nec-1s. As an important downstream effector of necroptosis, MLKL is known to cause cell membrane rupture, which can be assessed by LDH release. We found that LDH release into media was dramatically increased by LPS/QVD/BV6 treatment compared to that observed when treated with LPS alone, and Nec-1/Nec-1s effectively reduced the LDH release (Figure 2C). The data suggest that Nec-1/Nec-1s may suppress inflammation by blocking the DAMPs released from necroptotic cells [12,29]. In accordance with this, we found that Nec-1/Nec-1s efficiently suppressed high mobility group box1 (HMGB1) release, which was dramatically increased in LPS/QVD/BV6-treated cells (Figure 2D). Nec-1/Nec-1s also inhibited HMGB1 expression at the protein and mRNA levels in necroptotic cells (Figure 2E,F). Furthermore, Nec-1/Nec-1s suppressed the expression of other DAMP molecules, such as IL-1α, IL-1β, and IL-33 (Figure 2F).

### 3.4. Nec-1/Nec-1s Exerted Anti-Inflammatory Effects by Modulating AMPK, PI3K/Akt, MAPKs, and NF-κB Signaling Pathways in LPS-Stimulated BV2 Cells

RIPK1 not only plays a role as a key regulator of necroptotic cell death but also plays a central role in regulating innate immunity and inflammation [15,16]. Thus, we analyzed the anti-inflammatory mechanism of Nec-1/Nec-1s under LPS-induced inflammatory conditions. Western blot and RT-PCR analyses showed that Nec-1 and Nec-1s reduced the expression of inducible nitric oxide synthase (iNOS), TNF-α, IL-1β, and IL-6 and increased IL-10 at mRNA and protein levels in LPS-stimulated BV2 cells (Figure 3A,B and Appendix A). Next, we examined the effect of Nec-1/Nec-1s on MAP kinases and Akt, which are important signaling molecules mediating pro-inflammatory gene expression [27]. Nec-1 and Nec-1s inhibited LPS-induced phosphorylation of ERK, p38MAPK, and Akt, but not JNK (Figure 3C,D and Appendix A). In contrast, Nec-1/Nec-1s enhanced LPS-induced phosphorylation of AMPK, which is a key anti-inflammatory signaling molecule in microglia [31,32] (Figure 3E and Appendix A). Finally, we demonstrated that Nec-1/Nec-1s inhibited the DNA binding, nuclear translocation, and transcriptional activity of NF-κB in LPS-stimulated BV2 cells (Figure 3F–H and Appendix A).

### 3.5. Nec-1/Nec-1s Exerted Antioxidant Effects by Modulating Nrf2/ARE and PKA/CREB Signaling Pathways in LPS-Stimulated BV2 Cells

ROS are known to be early signaling inducers of inflammatory reactions [33] and stimulate RIPK1 auto-phosphorylation and contribute to RIPK3 recruitment into the necrosome [34,35]. Moreover, RIPK1 is known to trigger mitophagy and ROS production [36]. In the present study, we found that Nec-1 and Nec-1s significantly reduced intracellular ROS levels in LPS-stimulated BV2 cells (Figure 4A and Appendix A). Using qRT-PCR analysis, we found that Nec-1/Nec-1s inhibited ROS production by suppressing the mRNA expression of p22phox among NADPH oxidase subunits (Figure 4B and Appendix A). To further analyze the antioxidant mechanism, we examined the effects of RIPK1 inhibitors on nuclear factor-erythroid 2 (NF-E2)-related factor (Nrf2)/ARE and CREB signaling. Nec-1/Nec-1s increased the antioxidant enzyme HO-1 expression and its upstream transcription factor Nrf2 DNA binding activity in LPS-stimulated BV2 cells (Figure 4C–E and Appendix A). In addition, Nec-1/Nec-1s increased the reporter gene activity of ARE harboring the Nrf2 binding site (Figure 4F and Appendix A). Furthermore, Nec-1 and Nec-1s increased the phosphorylation, nuclear translocation, and reporter gene activity of CREB in LPS-stimulated BV2 cells (Figure 4G–I and Appendix A).

### 3.6. Nec-1 and Nec-1s Inhibited Microglial Activation and Proinflammatory Gene Expression in the Brains of LPS-Injected Mice

To confirm the anti-inflammatory effect of Nec-1 and Nec-1s in vivo, mice were administered Nec-1 or Nec-1s before LPS injection. After 24 h, the mice were sacrificed, and microglial activation was assessed by staining their brain tissue with an antibody against Iba-1, a marker of microglial activation. LPS treatment increased the number of Iba-1-positive activated microglial cells, whereas Nec-1or Nec-1s treatment decreased the number of activated microglia cells in the dentate gyrus (DG), cerebral cortex, and substantia nigra (SN) (Figure 5A,B; DG, F_3, 65_ = 33.12, *p* < 0.01; cortex, F_3, 64_ = 88.63, *p* < 0.01; SN, F_3, 63_ = 185.15, *p* < 0.01). When we examined the effect of Nec-1 on gene expression in LPS-injected mouse brains, Nec-1 inhibited the expression of iNOS, pro-inflammatory cytokines (TNF-α, IL-1β, IL-6), matrix metalloproteinases (MMPs; MMP-3, -8), TLR2, and Iba-1, and increased the anti-inflammatory cytokine IL-10 and antioxidant enzyme HO-1 (Figure 5C,D; iNOS, F_3, 8_ = 34.00, *p* < 0.01; TNF-α, F_3, 8_ = 48.59, *p* < 0.01; IL-1β, F_3, 8_ = 324.51, *p* < 0.01; IL-6, F_3, 8_ = 1225.62, *p* < 0.01; MMP-3, F_3, 8_ = 1182.96, *p* < 0.01; MMP-8, F_3, 8_ = 127.17, *p* < 0.01; TLR2, F_3, 8_ = 24.06, *p* < 0.01; Iba-1, F_3, 8_ = 46.75, *p* < 0.01; IL-10, F_3, 8_ = 19.18, *p* < 0.01; HO-1, F_3, 8_ = 41.21, *p* < 0.01). Moreover, Nec-1 suppressed LPS-induced expression of DAMP molecules such as HMGB1 and IL-1α (Figure 5C,D; HMGB1, F_3, 8_ = 72.58, *p* < 0.01; IL-1α, F_3, 8_ = 101.09, *p* < 0.01).

### 3.7. Nec-1/Nec-1s Inhibited the Phosphorylation and Expression of RIPK1-RIPK3-MLKL in the Brains of LPS-Injected Mice

To investigate whether necroptosis occurred in LPS-induced inflammatory responses in vivo, we examined the expression of necroptosis markers in LPS-injected mouse brains. Western blot analysis showed that LPS treatment increased the phosphorylation and expression of RIPK1, RIPK3, and MLKL proteins in the cortex, which were inhibited by Nec-1 (Figure 6A; pRIPK1, F_3, 8_ = 61.80, *p* < 0.01; pRIPK3, F_3, 8_ = 9.18, *p* < 0.01; pMLKL, F_3, 8_ = 19.79, *p* < 0.01; RIPK1, F_3, 8_ = 64.44, *p* < 0.01; RIPK3, F_3, 8_ = 14.89, *p* < 0.01; MLKL, F_3, 8_ = 6.78, *p* < 0.05). In addition, Nec-1 inhibited the mRNA expression of RIPK1, RIPK3, and MLKL induced by LPS (Figure 6B; RIPK1, F_3, 8_ = 86.68, *p* < 0.01; RIPK3, F_3, 8_ = 34.80, *p* < 0.01; MLKL, F_3, 8_ = 40.69, *p* < 0.01). To confirm the expression of p-RIPK1 in microglia, we performed co-immunostaining using antibodies against OX-42 (microglial marker) and RIPK1 phosphorylated at serine 166, a marker for autophosphorylation and activation [18,37]. As a result, LPS increased the p-RIPK1 expression in microglia, which was decreased by Nec-1/Nec-1s in the mouse brains (Figure 6C; pRIPK1^+^, F_3, 18_ = 11.01, *p* < 0.01; pRIPK1^+^/OX-42^+^, F_3, 18_ = 7.61, *p* < 0.01).

### 3.8. Nec-1 and Nec-1s Exerted Neuroprotective and Anti-Inflammatory Effects in MPTP-Induced PD Mice

We next investigated whether Nec-1/Nec-1s has neuroprotective effects in an acute model of MPTP-induced PD mice. Treatment of Nec-1 or Nec-1s recovered MPTP-induced dopaminergic neuronal cell death in the SN and striatum as demonstrated by TH staining (Figure 7A,B; striatum, F_3, 20_ = 38.82, *p* < 0.01; SN, F_3, 20_ = 18.42, *p* < 0.01). Biochemical analysis revealed that Nec-1 and Nec-1s restored the protein expression of TH and neurotrophic factors such as BDNF, GDNF, and PGC-1α, which were reduced by MPTP treatment (Figure 7C,D; TH, F_5, 12_ = 19.95, *p* < 0.01; BDNF, F_5, 18_ = 6.45, *p* < 0.01; GDNF, F_5, 12_ = 3.99, *p* < 0.05; PGC-1α, F_5, 12_ = 6.98, *p* < 0.01). Moreover, Nec-1 and Nec-1s suppressed microglial activation and the expression of proinflammatory cytokines such as TNF-α, IL-1β, and IL-6 in MPTP-injected mouse brains (Figure 7E–H; striatum, F_3, 28_ = 61.25, *p* < 0.01; SN, F_3, 24_ = 926.99, *p* < 0.01; TNF-α, F_5, 12_ = 112.57, *p* < 0.01; IL-1β, F_5, 18_ = 8.27, *p* < 0.01; IL-6, F_5, 12_ = 23.09, *p* < 0.01).

### 3.9. Nec-1/Nec-1s Reduced p-RIPK1 Expression in Microglia of MPTP-Induced PD Mice

To identify the cell types that express p-RIPK1 in MPTP-induced PD mice, we performed co-immunostaining using antibodies against p-RIPK1 (Ser 166), OX-42, or TH. As a result, p-RIPK1 was increased by MPTP and then decreased by Nec-1 and Nec-1s in the SN (Figure 8A; pRIPK1^+^, F_5, 45_ = 9.78, *p* < 0.01). We observed that p-RIPK1-positive cells colocalized with OX-42-positive cells but not with TH-positive cells in the SN (Figure 8A,B; pRIPK1^+^/OX-42^+^, F_5, 36_ = 28.57, *p* < 0.01; pRIPK1^+^/TH^+^, F_5, 35_ = 0.73, *p* > 0.05). The data suggest that p-RIPK1 is mainly expressed in microglia and may contribute to neuroinflammation, which in turn leads to dopaminergic cell death in MPTP mice. Nec-1 and Nec-1s may recover dopaminergic neuronal cell death by attenuating inflammation by regulating RIPK1 activity in the microglia.

## 4. Discussion

In the present study, we demonstrated the role of RIPK1 in neuroinflammation and PD mouse models. The inhibition of RIPK1 by Nec-1/Nec-1s suppressed proinflammatory cytokines and cell death in LPS-induced inflammatory conditions and LPS/QVD/BV6-induced necroptotic conditions of BV2 microglial cells. The Nec-1/Nec-1s also suppressed microglial activation and the expression of proinflammatory and necroptotic markers in the brains of LPS-injected mice. Furthermore, Nec-1/Nec-1s protected against dopaminergic neuronal cell death and inhibited microglial activation in the MPTP-induced PD mouse model. Intriguingly, MPTP increased the p-RIPK1 expression in microglia but not in TH^+^ neurons in MPTP mice, whereas Nec-1/Nec-1s decreased the p-RIPK1 expression in microglia. The findings suggest that RIPK1 may mediate neuroinflammation, which leads to neuronal cell death in this acute PD model mice.

Nec-1, the first small-molecule inhibitor of RIPK1, was isolated in a cell-based high-throughput chemical screen for inhibitors of caspase-independent necrosis. Nec-1s is an improved analog of Nec-1 and is a highly specific inhibitor of both human and murine RIPK1 with excellent blood–brain barrier permeability and reasonable oral availability [16]. Because Nec-1 and Nec-1s can effectively inhibit both human and murine RIPK1, they have been widely used to investigate the role of RIPK1 and necroptosis in various cellular and animal models of human diseases [38]. Given that Nec-1 also inhibits the potent immunomodulatory enzyme indolamine 2,3-dioxygenase (IDO) [39], the anti-inflammatory effect of Nec-1 in this study could possibly be due to decreased IDO. However, the results of Nec-1 are nearly identical to those of Nec-1s (which do not inhibit IDO), suggesting that the anti-inflammatory effects of Nec-1 are RIPK1-specific.

Recently, RIPK1 has emerged as a key mediator of neuroinflammation [16,40]. The RIPK1 increase has been observed in the microglia of AD, ALS, and MS mouse models, and Nec-1 treatment reduced proinflammatory cytokine expression [18,19,41]. In addition, RIPK1-mediated neuroinflammation has been suggested to provoke Aβ-like protein aggregation and neurodegeneration [16,18]. Activated RIPK1 and RIPK3 lead to the formation of amyloid-like fibrils and thus may act as a seeding mechanism of protein aggregation. Therefore, the possibility is high that RIPK1-mediated neuroinflammation promotes protein misfolding and aggregation in many neurodegenerative diseases. In addition to the role of RIPK1 in the necroptosis pathway, RIPK1 has non-necroptotic functions, such as RIPK1-dependent apoptosis and regulation of pro-inflammatory gene expression [9,42]. For example, the RIPK1 inhibitor (Nec-1s) inhibited LPS-induced inflammation via ERK, c-Fos, and NF-κB in the absence of necroptosis [42]. Thus, RIPK1 inhibition has been suggested as a prime therapeutic target for neuroinflammation and neurodegeneration. However, the role of RIPK1 and its inhibitors in microglial activation has not been clearly demonstrated.

In the present study, we analyzed the anti-inflammatory effect and mechanism of the RIPK1 inhibitor in LPS-stimulated BV2 microglial cells (in this situation, necroptosis occurs minimally, if any, as shown by LDH assay data). Both Nec-1 and Nec-1s suppressed the expression of proinflammatory cytokines and increased the anti-inflammatory IL-10 at mRNA and protein levels. Subsequently, we found that Nec-1/Nec-1s exert anti-inflammatory effects by inhibiting the phosphorylation of ERK, p38MAPK, and Akt with the enhancement of AMPK phosphorylation, and by reducing the NF-κB activity in LPS-stimulated BV2 cells. Moreover, Nec-1/Nec-1s showed antioxidant effects by upregulating Nrf2/HO-1 and CREB signaling. AMPK has been shown to play a central role as an anti-inflammatory mediator in microglia [31,32]. The activation of AMPK signal pathways has been demonstrated to promote M2 macrophage/microglia polarization, thereby inhibiting inflammation [43,44]. Our group recently reported that AMPK activation mediates anti-inflammatory effects by upregulating PKA/CREB, Nrf2/ARE signaling and downregulating proinflammatory signals such as ROS, PI3K/Akt, p38MAPK, and NF-κB [32]. Therefore, our results suggest that Nec-1/Nec-1s exert an anti-inflammatory effect mainly by activating AMPK and its downstream signaling molecules in LPS-stimulated microglia. Further studies are necessary to determine whether the same mechanisms are applicable in in vivo microglial activation.

The increase of necroptosis and necroptotic-related proteins have been reported in the SN of PD patients and MPTP- or 6-hydroxydopamine (6-OHDA)-induced PD mice [22,23,45]. Additionally, the neuroprotective effects of necroptosis inhibition have been reported in those models. In MPTP mouse models, Nec-1/Nec-1s/miR-425 decreased RIPK1/3 and MLKL overexpression and activation and restored striatal dopamine loss and nigral dopaminergic neuronal death [21,22,45]. In addition, RIPK3 or MLKL knockout mice showed similar neuroprotective effects. In the 6-OHDA mouse model, Nec-1s or RIPK3/MLKL deficiency attenuated necroptotic-driven axonal degeneration in the striatum and retrograde nigral dopaminergic neurodegeneration [23]. However, the detailed mechanisms underlying the effect of necroptotic factors and their inhibitors on dopaminergic neuronal death/survival and neuroinflammation remain unclear. Therefore, the present study investigated the anti-inflammatory/neuroprotective effects of RIPK1 inhibitors and their underlying molecular mechanisms in MPTP-induced PD mice. The Nec-1/Nec-1s recovered dopaminergic neuronal cell death by upregulating the expression of neurotrophic factors such as BDNF, GDNF, and PGC1α. In addition, Nec-1/Nec-1s showed anti-inflammatory effects by inhibiting pro-inflammatory cytokines such as TNF-α, IL-1β, and IL-6. Interestingly, Nec-1/Nec-1s reduced p-RIPK1 expression and its colocalization with microglia in the SN of MPTP mice. The findings imply that RIPK1 may play a critical role in neuroinflammation and promote harmful inflammatory signaling, which provides a feedforward mechanism to induce neurodegeneration.

## 5. Conclusions

To the best of our knowledge, this study is the first to report on the role of RIPK1 as a key regulator of microglial activation under in vitro and in vivo neuroinflammatory conditions. The RIPK1 inhibitors Nec-1/Nec-1s showed anti-inflammatory and neuroprotective effects in LPS-induced neuroinflammation and MPTP-induced PD mouse models. Therefore, controlling RIPK1 is a promising therapeutic strategy for neuroinflammatory and neurodegenerative disorders, including PD.

## Figures and Tables

**Figure 1 cells-12-00417-f001:**
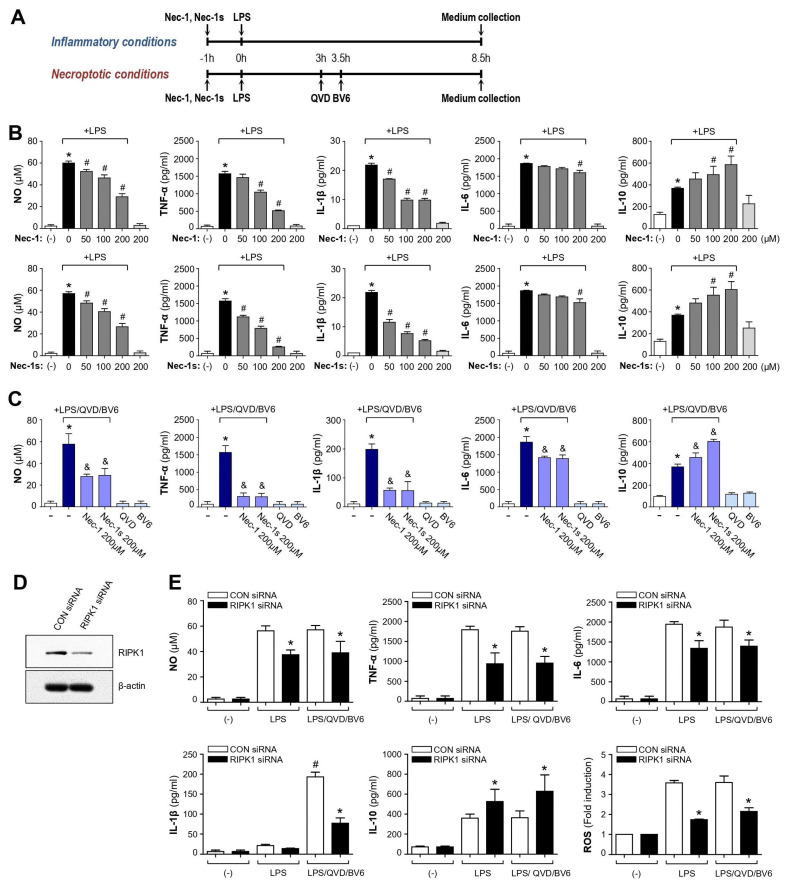
Effect of Nec-1, Nec-1s, and RIPK1 siRNA on NO and inflammatory molecules in LPS or LPS/QVD/BV6-stimulated BV2 microglial cells. (**A**) Scheme of the experimental procedure. (**B**) BV2 cells were pretreated with Nec-1 or Nec-1s for 1 h and incubated with LPS (100 ng/mL). After incubation for 8.5 h, the supernatants were obtained, and the levels of NO, TNF-α, IL-1β, IL-6, and IL-10 were measured. (**C**) BV2 cells were pretreated with Nec-1 and Nec-1s for 1 h and preincubated with or without LPS for 3 h and were then pretreated with QVD (10 μM) for 30 min before stimulation with BV6 (5 μM) for 5 h. The levels of NO and cytokines released into media were measured. All data are shown as the mean value ± SEM of three independent experiments. * *p* < 0.05 vs. control; ^#^
*p* < 0.05 vs. LPS-treated samples; ^&^
*p* < 0.05 vs. LPS/QVD/BV6-treated samples. (**D**) Downregulation of RIPK1 expression by RIPK1 siRNA was confirmed by Western blot analysis. (**E**) The cells transfected with RIPK1 siRNA or control siRNA were treated with LPS or LPS/QVD/BV6 for 8.5 h. Then, the amounts of cytokines released into the media and intracellular ROS levels were determined. All data are shown as the mean value ± SEM of three independent experiments. ^#^
*p* < 0.05 vs. CON siRNA-transfected cells in the presence of LPS, * *p* < 0.05 vs. CON siRNA-transfected cells in the presence of LPS or LPS/QVD/BV6.

**Figure 2 cells-12-00417-f002:**
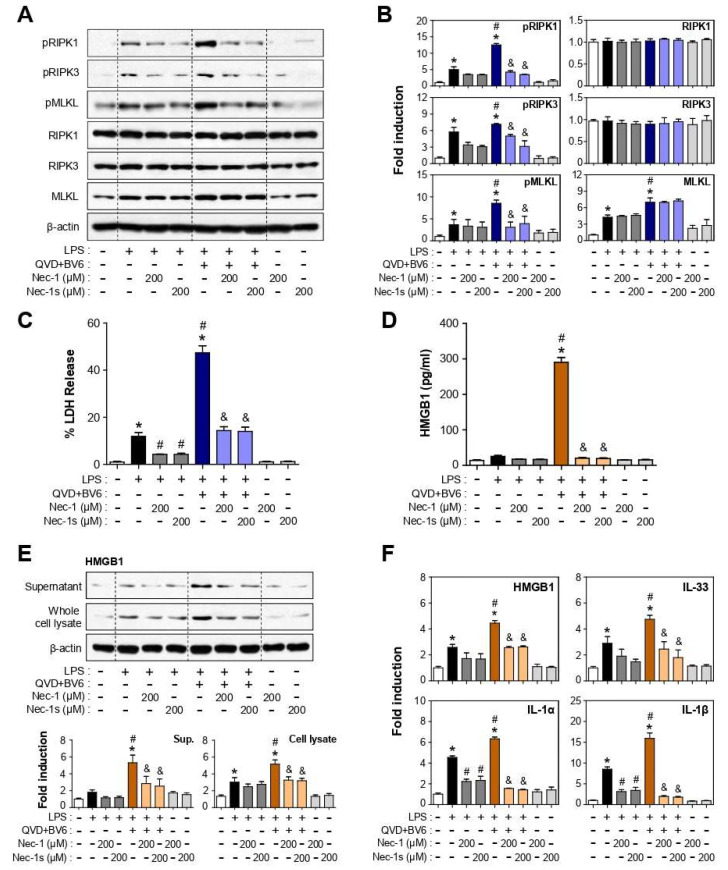
Effect of Nec-1/Nec1-s on the phosphorylation of RIPK1-RIPK3-MLKL, cell death, and the expression of DAMPs in LPS and LPS/QVD/BV6-stimulated BV2 cells. (**A**,**B**) Cell lysates were obtained from LPS or LPS/QVD/BV6-treated cells in the absence or presence of Nec-1/Nec-1s for 8.5 h and analyzed by Western blot for expression of phospho- and total form of RIPK1, RIPK3, MLKL. Representative gels are shown in the left panel, and the quantification of three experiments is shown in the right panel. (**C**,**D**) Culture supernatants were collected from aforementioned treated cells to determine the level of LDH (**C**) and HMGB1 (**D**) release. (**E**) Western blot data showing the effect of Nec-1/Nec-1s on HMGB1 protein expression in cell lysates and supernatant. (**F**) Quantitative real-time PCR data showing the effect of Nec-1/Nec-1s on mRNA expression of DAMPs; HMGB1, IL-1α, IL-1β, IL-33. All data are shown as the mean value ± SEM of three independent experiments. * *p* < 0.05 vs. control; ^#^
*p* < 0.05 vs. LPS-treated samples; ^&^
*p* < 0.05 vs. LPS/QVD/BV6-treated samples.

**Figure 3 cells-12-00417-f003:**
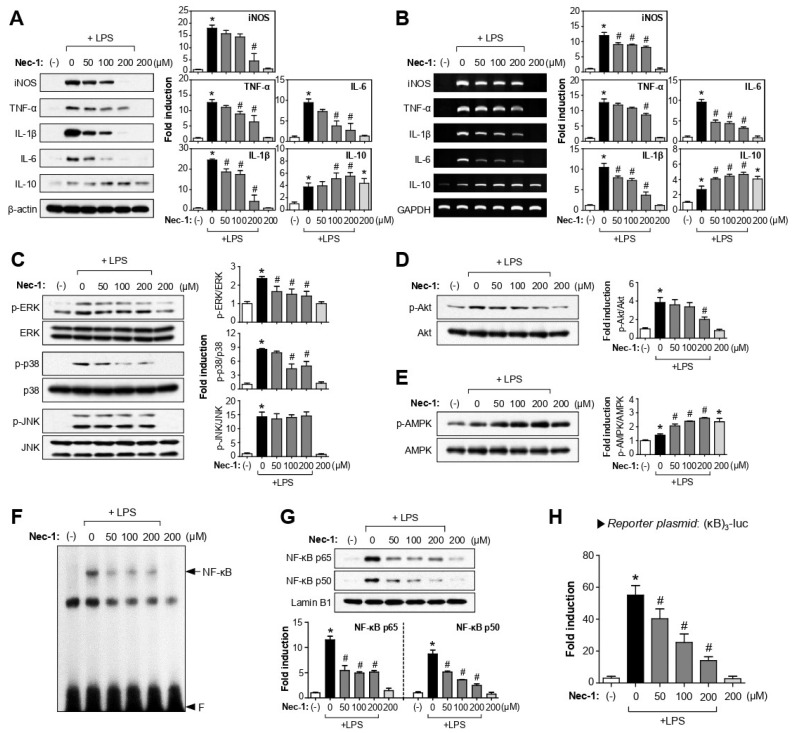
Effect of Nec-1 on inflammatory cytokines and signaling molecules such as MAPKs, Akt, AMPK, and NF-κB in LPS-stimulated BV2 cells. (**A**,**B**) BV2 cells were pretreated with Nec-1 for 1 h and incubated with LPS (100 ng/mL) for 6 h. Western blot analysis (**A**) and RT-PCR (**B**) were performed to measure the expression of TNF-α, IL-1β, IL-6, and IL-10. (**C**–**E**) BV2 cells were pretreated with Nec-1 and incubated with LPS (100 ng/mL) for 1 h. Western blot analysis using antibodies against the phospho-forms of ERK, p38, JNK, Akt, and AMPK were normalized with respect to the level of each total form and expressed as fold changes relative to the control group. (**A**–**E**), representative gels are shown in the left panel, and the quantification of three independent experiments is shown in the right panel. (**F**) EMSA for NF-κB was performed using nuclear extracts prepared from BV2 cells pretreated with Nec-1 and incubated with LPS (100 ng/mL) for 1 h. (**G**) Effect of Nec-1 on nuclear translocation of NF-κB subunits. (**H**) Transient transfection analysis of (κB)_3_-luc reporter gene activity. Data are shown as the mean ± SEM of three independent experiments. * *p* < 0.05 vs. control; ^#^
*p* < 0.05 vs. LPS-treated samples.

**Figure 4 cells-12-00417-f004:**
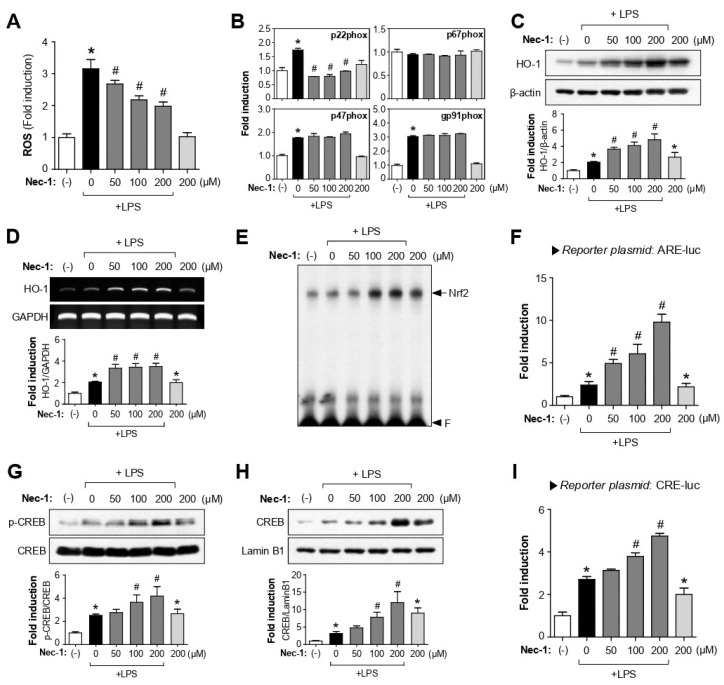
Nec-1 reduced ROS production via suppression of p22phox NADPH oxidase subunit and upregulation of Nrf2/ARE and PKA/CREB signaling. (**A**) BV2 cells were treated with Nec-1 1 h prior to LPS stimulation for 8.5 h. Intracellular ROS level was measured using the DCF-DA assay. (**B**) Quantitative real-time PCR to assess mRNA expression of NADPH oxidase subunits (p22phox, p47phox, p67phox, and gp91phox) in BV2 cells. (**C**,**D**) Western blot and RT-PCR analysis to determine the effect of Nec-1 on the protein and mRNA expression of HO-1. BV2 cells were treated with Nec-1 for 1 h, followed by LPS (100 ng/mL) for 6 h. (**E**) EMSA for Nrf2 DNA binding activity. BV2 cells pretreated with Nec-1 and incubated with LPS (100 ng/mL) for 1 h. (**F**) Transient transfection analysis of ARE-luc reporter gene activity. (**G**,**H**) Western blot analysis for p-CREB and total CREB was performed using cell lysates and nuclear extracts from BV2 cells, respectively. (**I**) Transient transfection analysis of CRE-luc reporter gene activity. Data are shown as the mean ± SEM of three independent experiments. * *p* < 0.05 vs. control; ^#^
*p* < 0.05 vs. LPS-treated samples.

**Figure 5 cells-12-00417-f005:**
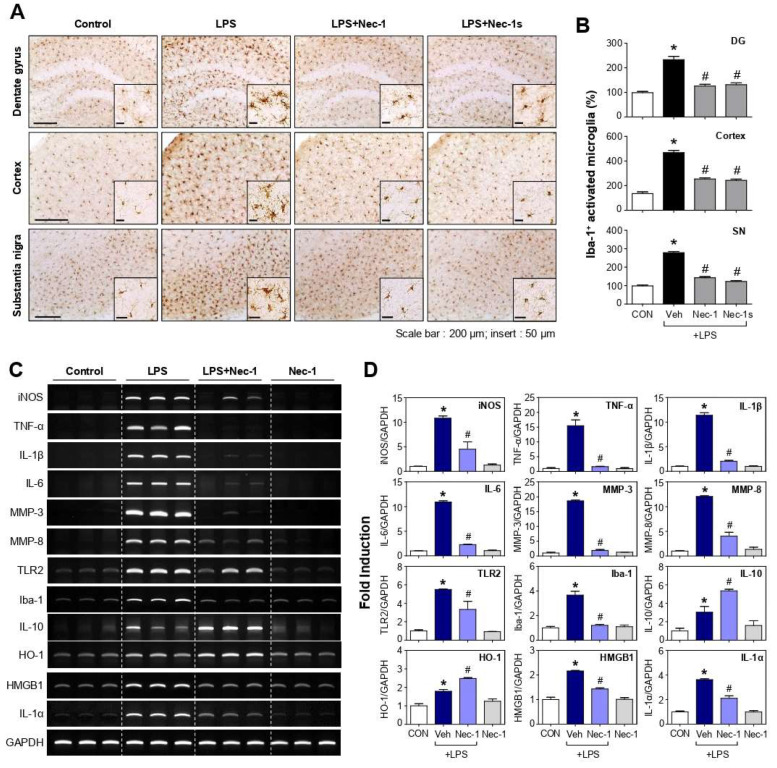
Effects of Nec-1 and Nec-1s on microglial activation and pro-/anti-inflammatory markers in the brains of LPS-injected mice. (**A**) Immunohistochemical staining for Iba-1 (microglial marker) and (**B**) quantification of the number of Iba-1-positive activated microglia 24 h after LPS injection (number of mice in each group *n* = 4–5). Microglial activation in the dentate gyrus (DG), cortex and substantia nigra (SN). Scale bars, 200 μm; insert, 50 μm (**C**) Effects of Nec-1 on the mRNA levels of iNOS, pro-inflammatory cytokines (TNF-α, IL-1β, IL-6), MMPs (MMP-3, MMP-8), TLR2, Iba-1, IL-10, HO-1, and DAMPs (HMGB1, IL-1α) in the cortex of LPS-injected mice. (**D**) Quantification data are shown. Data are shown as the mean ± SEM. * *p* < 0.05 vs. control group; ^#^
*p* < 0.05 vs. LPS group.

**Figure 6 cells-12-00417-f006:**
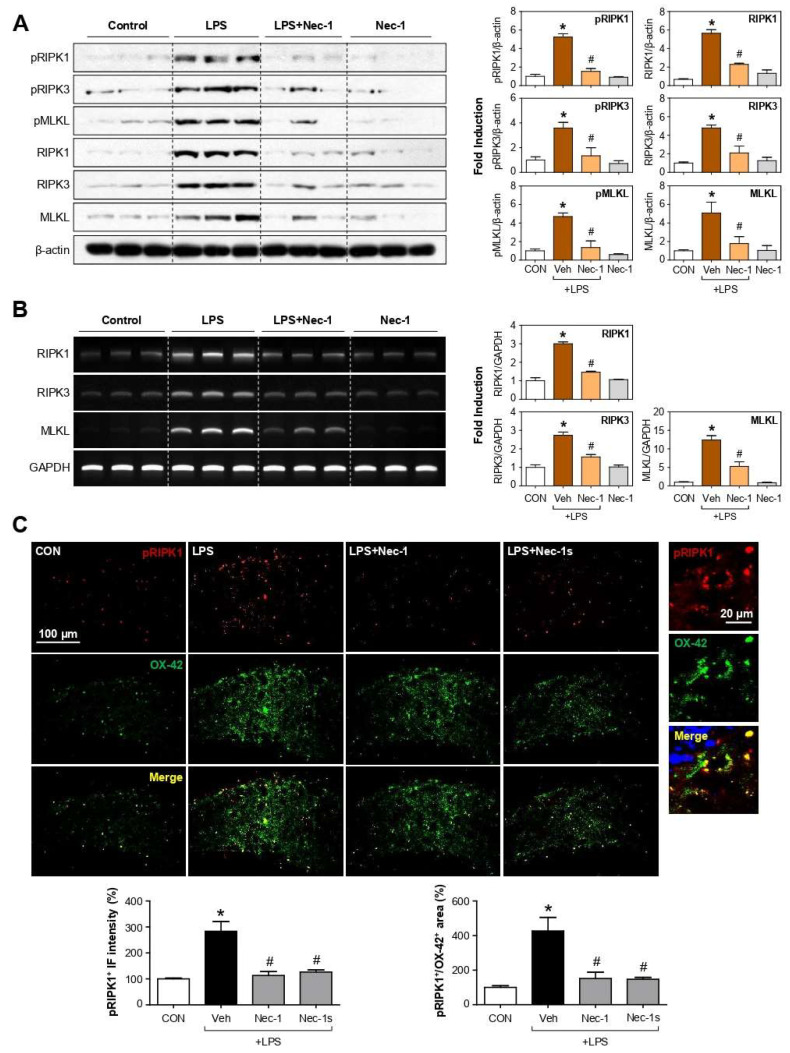
Effects of Nec-1 on protein and mRNA expression of necroptosis markers in the brains of LPS-injected mice. (**A**) Western blot analysis using antibodies against the phospho- or total forms of RIPK1, RIPK3, and MLKL in the cortex of LPS-injected mice. (**B**) RT-PCR was performed to measure the expression of RIPK1, RIPK3, and MLKL mRNA. Representative gels are shown in the left panel and quantifications in the right panel. (**C**) Double immunofluorescence (IF) staining for p-RIPK1 ser166 (red, auto-phosphorylated RIPK1 marker) and OX-42 (green, microglial marker) in DG of LPS-injected mice. Representative images are shown. Quantification of the p-RIPK1 positive intensity (%) is shown in the left panel, and p-RIPK1/OX-42 positive colocalization (%) is shown in the right panel. Data are shown as the mean ± SEM. * *p* < 0.05 vs. control group; ^#^
*p* < 0.05 vs. LPS group.

**Figure 7 cells-12-00417-f007:**
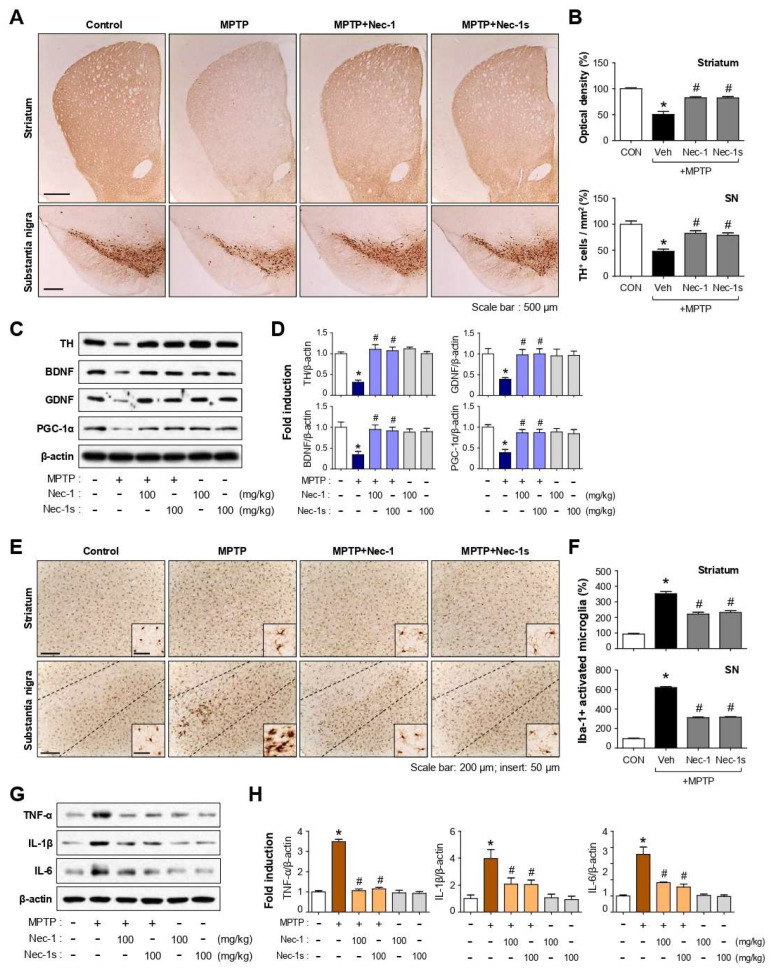
Effect of Nec-1 and Nec-1s on dopaminergic neuronal cell death, microglial activation in the brains of MPTP-injected mice. (**A**) Representative images of TH-positive neuronal cells in the striatum and SN (each group *n* = 4–5). (**B**) Quantitative analysis was performed by measuring the optical density of TH-positive fibers in the striatum and the number of TH-positive cells in the SN. (**C**) Western blot data showing the effect of Nec-1/Nec-1s on the expression of neurotrophic factors in the SN. (**D**) Quantification of Western blot data. (**E**) Immunohistochemical staining for Iba-1 (microglial marker) in striatum and SN, and (**F**) quantification of the number of Iba-1-positive-activated microglia in the striatum and SN. (**G**) Western blot data shows the effect of Nec-1/Nec-1s on the expression of proinflammatory cytokines in the SN. (**H**) Quantification of Western blot data. Data are shown as the mean ± SEM. * *p* < 0.05 vs. control group; ^#^
*p* < 0.05 vs. MPTP group.

**Figure 8 cells-12-00417-f008:**
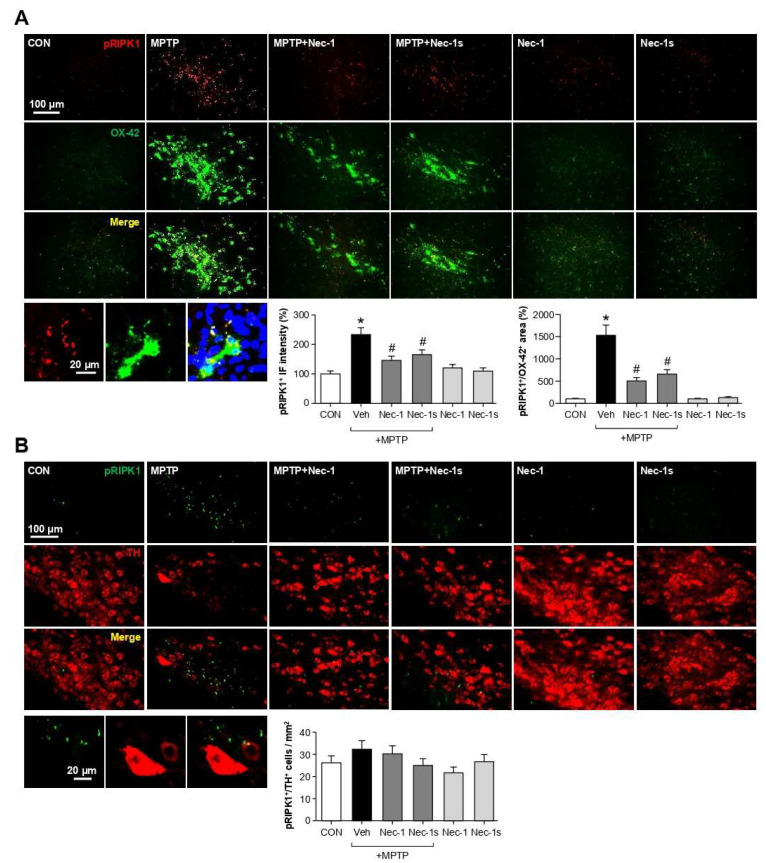
Effect of Nec-1 and Nec-1s on p-RIPK1 (Ser 166) in the brains of MPTP-injected mice. (**A**) Double IF staining for p-RIPK1 Ser 166 (red) with OX-42 (green, microglial marker) in the SN of MPTP-injected mice. Representative images are shown in the upper panel, and quantifications of p-RIPK1 positive intensity and p-RIPK1/OX-42 positive colocalization are shown in the bottom panel. (**B**) Double IF staining for p-RIPK1 Ser 166 (green) and TH (red, dopaminergic neuronal marker) in the SN of MPTP-injected mice. Representative images are shown in the upper panel, and quantification data of the p-RIPK1/TH positive colocalization is shown in the bottom panel. Data are shown as the mean ± SEM. * *p* < 0.05 vs. control group; ^#^
*p* < 0.05 vs. MPTP group.

**Table 1 cells-12-00417-t001:** Primer sequences used for PCR.

Gene	Forward Primer (5′-3′)	Reverse Primer (5′-3′)
*iNOS*	CAAGAGTTTGACCAGAGGACC	TGGAACCACTCGTACTTGGGA
*TNF-* *α*	CCTATGTCTCAGCCTCTTCT	CCTGGTATGAGATAGCAAAT
*IL-1α*	GACCAGCCCGTGTTGC	AGTCCCCGTGCCAGGT
*IL-1β*	GATCCACACTCTCCAGCTGCA	CAACCAACAAGTGATATTCTCCATG
*IL-6*	CCACTTCACAAGTCGGAGGCTT	CCAGCTTATCTGTTAGGAGA
*IL-10*	GCCAGTACAGCCGGGAAGACAATA	GCCTTGTAGACACCTTGGTCTT
*HMGB1*	GCAAAGAAACTAGGAGAGAT	TCTTCTTCATCTTCGTCTTC
*IL-33*	GAATTCTGCCATGTCTACTG	CCTTGGATGCTCAATGTG
*MMP-3*	CTCCAGTATTTGTCCTCTAC	TGGAACCACTCGTACTTGGGA
*MMP-8*	CCAAGGAGTGTCCAAGCCAT	CCTGGTATGAGATAGCAAAT
*TLR2*	TGCTTTCCTAGCTGGAGATTT	AGTCCCCGTGCCAGGT
*Iba-1*	AGGAGATTTCAAAAGCTGATGTGG	CAACCAACAAGTGATATTCTCCATG
*HO-1*	ATACCCGCTACCTGGGTGAC	CCAGCTTATCTGTTAGGAGA
*p22phox*	CAATGGCCAAGCAGACGGTC	GCCTTGTAGACACCTTGGTCTT
*p47phox*	GTTTCAGGTCATCAGGCCGC	TCTTCTTCATCTTCGTCTTC
*p67phox*	AGGCCACTGCAGAGTGCTTG	CCTTGGATGCTCAATGTG
*gp91phox*	TGGCGGTGTGCAGTGCTATC	ATTCAGTCCCTCTATGGA
*RIPK1*	AGGGTCATGCAGTTTGGAAC	CCTGCAGGAAAACTGCATCG
*RIPK3*	ATGTCCTGAGAGGCAAGCAC	TGTAACGCAACAGCTTCAGG
*MLKL*	CCCATTTGAAGGCTGTGATT	GTTTGGACGGCAGATCCTCA
*GAPDH*	GGCATGGACTGTGGTCATGA	TGTCACCCTGTGCTTGACCT

## Data Availability

The datasets generated for this study are available on request to the corresponding authors.

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
