# Peer review of "RIPK1 Regulates Microglial Activation in Lipopolysaccharide-Induced Neuroinflammation and MPTP-Induced Parkinson’s Disease Mouse Models"

_cells, 2023, doi:10.3390/cells12030417_

Round 1

Reviewer 1 Report

In this article entitled “RIPK1 regulates microglial activation in lipopolysaccharide-induced neuroinflammation and MPTP-induced Parkinson’s disease mouse models” the authors detailed the mechanisms of RIPK1 in two models. The corresponding has published studies evaluating other agents such as  phosphodiesterase 10 inhibitor, antioxidants NQO1 following similar methodologies in models of neuroinflammation. The anti-inflammatory effects of nec-1 and nec-1s have been reported by others in MS, AD and models of neuroinflammation. In the present manuscript the authors use the traditional methods in evaluating the mechanisms of RIPK1 additionally in a PD model.

Although the experiments are well designed and interpreted, there is not much of new information in the manuscript except for the effects of Nec-1/Nec-1s in the PD model. Why was OX42 preferred as a microglial marker rather than other more specific marker such as IBA1.

Some part of the “Introduction” section (Page 2 lines 53-61) and the “discussion” section (Page 17, lines 490-493) should be edited and modified for better clarity.

Reviewer 2 Report

The study by Do-Yeon Kim and colleagues investigates the role of RIPK1 in microglial activation using different models. The manuscript is well written, the results are clear and interpretations are scientifically relevant. However, to convey better the hypothesis and scientific questions, the introduction section needs more work, considering that this is a journal with wide scope and readership. I would strongly suggest to address some of my comments below to increase the quality and impact of this work:

-          Introduction, there are several proteins mentioned in this section that are written with acronyms. This is ok for most proteins with long history and background like TNF of NF-kb for instance. But the authors should give more detail or at least spell out the ones are less known i.e., NEMO, CYLD, FADD, etc. Since this is a journal with a wide scope and diverse background readers it is strongly recommended to provide the full names somewhere in the manuscript, e.g. in a supplementary table.

-          The introduction section is very well elaborated in terms of the function and background of RIPK1 and Nec-1. However, current knowledge about the cellular and animal models proposed to address the scientific questions are not provided in this section, or at least they are barely mentioned. How LPS induces neuroinflammation? What is MPTP? Is this a robust and reliable model to demonstrate the hypothesis? Does the phenotype produced by MPTP match exactly with PD, or are there some side effects or caveats in using it?

-          Methods, line 205, better use “cryostat” instead of “cryotome”

-          Methods, 2.4, why the authors chose this LPS dose-time? Please cite previous work to support your rationale here.

-          Stats, I am a bit concern that using LSD as post hoc is not the most appropriate. LSD lacks correction of multiple comparisons, which is ok when working with max 3 experimental groups. For this work, where many groups are compared, I would suggest using a post hoc that considers correction of multiple comparisons

-          Fig. 2, Full membrane pictures should be provided as well, eg in a supplementary figure

Why in most of the figures, the control group bars do not display error bars? This is a typical mistake due to set control samples as 1. The calculation of the relative expression of control groups should be adjusted to convey their real variability. This can bring misinterpretation in comparisons against control groups that will increase the chances of type I error.

Reviewer 3 Report

Authors study the role of RIPK1 in the activation of microglia in LPS-induced neuroinflammation and MPTP-induced PD model of mice. Using RIPK1 inhibitors Nec-1 and Nec-1s, authors show a reduction in the levels of proinflammatory molecules and cell death and upregulation in the anti-inflammatory molecules in various models. Authors show the mechanism that Nec-1 and Nec-1s inhibit the phosphorylation of RIPK1 which is mainly expressed in microglia to mediate the anti-inflammatory effects. Since RIPK1 is mainly expressed in microglia, authors suggest that RIPK1 is a key regulator of microglial activation in the above neuroinflammation conditions. Overall, the study is impressive and well-performed. The mechanistic insight is compelling however several shortcomings need to be addressed before the study can be endorsed for publication.

I have the following major comments.

1.       Part of the study is conducted in BV2 cells in-vitro and the rest in-vivo. Especially the mechanistic part is largely based on the findings from the BV2 cells. Despite the similarities in these systems it simply cannot be claimed to be the same microglial mechanism in-vivo. Authors need to highlight this point in the discussion for an open interpretation.  

2.       A vast majority of the data relies on the reliable detection by antibodies therefore authors are recommended to provide specific details of antibodies, including catalog no., host, validation method or reference, and used concentrations.

3.       Result 3.1: the incubation duration of drugs is a bit confusing, so I request authors present a flow chart/illustration of the experimental design in figure 1.

4.       Result 3.6: the sample size seems to be 4-5; are these number of mice or the number of sections/images analyzed, please update.

5.       For in-vivo experiments, please mention in the statistics section whether the sample size was predetermined or not.

6.       Fig. 5 a: the morphological differences between and LPS group in the inserted images in DG are convincingly different however no such images are presented for the cortex and substantia nigra. Authors are requested to present them as well.

7.       Fig. 5 b: It is not clear whether authors counted the total number of cells in the field of view in all experimental groups or first categorized them as “activated” and counted only selected cells that showed this activated phenotype. In the case of counting the cells without any predefined criteria (as it appears from the control), why are they considered “activated” in control?

8.       Authors show no quantification of microglial morphology to claim them activated and a reduction in the numbers of activated microglial in the two treatment groups.

9.       Figure 6: the images in C are of poor resolution and the microglial marker is not showing the typical staining pattern of OX-42 to clearly see the colocalization at a larger population level. 

Round 2

Reviewer 2 Report

The authors addressed all my questions and comments. The paper should be accepted for publication